# FDTD Simulations of Shell Scattering in Au@SiO_2_ Core–Shell Nanorods with SERS Activity for Sensory Purposes

**DOI:** 10.3390/nano12224011

**Published:** 2022-11-15

**Authors:** Igor Kon, Andrey Zyubin, Ilia Samusev

**Affiliations:** REC «Fundamental and Applied Photonics. Nanophotonics», Immanuel Kant Baltic Federal University, A.Nevskogo 14, 236016 Kaliningrad, Russia

**Keywords:** FDTD, simulations, nanorods, SiO_2_ shell, optical sensor

## Abstract

The article describes the results of Finite-Difference Time-Domain (FDTD) mathematical modeling of electromagnetic field parameters near the surfaces of core–shell gold-based nanorods in the Au@SiO_2_ system. Three excitation linewidths (λ = 532, 632.8, and 785 nm) were used for theoretical experiments. Electric field parameters for Au nanorods, Au@SiO_2_ nanorods, and hollow SiO_2_ shells have been calculated and evaluated. The correlations between electric field calculated parameters with nanorod morphology and shell size parameters have been clarified. The optical properties of nanoobjects have been simulated and discussed. The highest maximum calculated value of the electric field tension was *E* = 7.34 V/m. The enhancement coefficient was ${\left|E/E_{0}\right|}^{4}$ = 3.15 × 10^7^ and was obtained on a rod with a SiO_2_ shell with dimensional parameters of height 70 nm, rod width 20 nm, and shell thickness 20 nm. As a result, a flexible simulation algorithm has been developed for the simulation of electric field parameters in each component of the Au@SiO_2_ system. The developed simulation algorithm will be applicable in the future for any other calculations of optical parameters in any similar component of the core–shell system.

## 1. Introduction

Gold nanoparticles (NPs) are currently widely used for modern sensory purposes [1,2,3]. Such NPs can be modified with various compounds and shells to obtain additional modalities. Gold NPs can be modified with antibodies [4] for targeted delivery [5], shells for encapsulating drugs [6], fluorescent and Raman labels to perform labeled fluorescence, and Raman spectroscopy [7]. Gold NPs are also used in the latest applications in physics and medicine, which represent the studies on the delivery of multimodal NPs to a target and simultaneously perform therapeutic, sensory, and targeted delivery functions [8]. In this case, the NP often has a shell that performs various functions of the structure. A coating shell can enhance colloidal stability, decrease toxicity, and allow for further functionalization of the NP to form a theranostic complex. Inorganic silica (SiO_2_) is widely used as a capsuling material due to its chemical inertness, optical transparency, and excellent shell thickness control [9]. Since Au NPs and Au@SiO_2_ NPs can act as energy coupling between free electrons and fluorophores, such structures can be applied for sensory purposes. The problem is to find the optimal shape of NPs and shells, in which the values of surface plasmon resonance will have the highest values. The former can be interpreted in the sphere of sensory functions of these NPs. Most of the papers focus mainly on forms of NPs [10] or represent the results of core–shell NPs morphological and optical properties investigation [11,12,13,14]. However, such works, especially those related to purely theoretical modeling, are not sufficient, especially those with a large comparative sample within geometrically identical NPs and their shells. There are a huge number of studies on surface-functionalized NPs dealing with their application to models of delivery systems, in which various parameters of multilayer NPs were indicated or slightly varied [15,16]. Optical processes involving Au NPs and NP-based complexes can be evaluated and described with simulation methods, such as the Finite-Difference Time-Domain (FDTD) method, for example [17,18]. Since silica shells can contribute to scattering, the FDTD method can be applied for calculations of NPs coated with silica shells [19]. Its optical properties and plasmon enhancement of the gold core can be calculated using FDTD.

This paper performs a theoretical calculation of electromagnetic fields near rod-shaped gold NPs for the gold nanorods (NRs) with/without shells, and with a single shell. We have investigated such optical parameters as the maximum value of the electric field, the integral sum of significant fields around the NPs, the recalculated values of the electric fields to the intensity of the Raman spectrum signal, scattering, and the theoretical Surface-Enhanced Raman scattering SERS enhancement factor. The applied problem includes, as noted above, NRs optimal sizes determination, shell thickness, and pure shell scattering. The thickness of the SiO_2_ layer around the NR, in this case, provides information about the behavior of plasmonic properties for such systems since it affects the electric field intensity near the surface of the NR. Scattering for a single (hollow) SiO_2_ shell was evaluated and compared with NR and SiO_2_@NR optical properties. The practical aspect of the study lies in the fundamental understanding of the rod shell contribution to the overall picture of the general optical properties in the case of the SiO_2_-coated rods. This, in turn, makes it possible to determine the optimal morphological parameters for the controlled synthesis of biocompatible nanoobjects with certain properties. In our previous paper, we simulate only spherical gold nanoparticles with a SiO_2_ shell system and obtain optimal dimensional characteristics for SERS experiments [18]. Another paper [17] describes the results of mathematical modeling of electric field strength distribution near the gold laser-induced periodic surface structures (LIPSS) without any shell.

## 2. Theoretical Model and Method

### 2.1. FDTD Approach

We used an approach based on the FDTD method using the basic Yee algorithm to solve Maxwell’s equations numerically. Yee proposed a spatial rectangular grid to discretize the selected computational domain [20]. The electric fields are located along the boundary of the Yee-described cube block [20], while the magnetic fields are located toward the center of the block. All grid components are spaced and independent of each other, thereby they will have different parameters at each point.

The fundamental Maxwell equations [Equation (1)]:(1)$$\frac{\partial \vec{B}}{\partial t}=-\vec{\nabla \;}\times \vec{E}-\vec{M},\;\frac{\partial \vec{D}}{\partial t}=\vec{\nabla \;}\times \vec{H}-\vec{J},\;\vec{\nabla \;}\vec{D}=0,\;\vec{\nabla \;}\vec{B}=0$$

For non-magnetic materials, the first two curl-equations have the following form [21] [Equation (2)]:(2)$$\frac{\partial \vec{D}}{\partial t}=\vec{\nabla \;}\times \vec{H},\;\vec{D}\left(\omega \right)=\varepsilon_{0}\varepsilon_{r}\left(\omega \right)\vec{E}\left(\omega \right),\;\frac{\partial \vec{H}}{\partial t}=-\frac{1}{\mu_{0}}\vec{\nabla \;}\times \vec{E}$$
where $\vec{H}$, $\vec{E}$ and $\vec{D}$ denote magnetic, electric field, and offset field, respectively, and $\varepsilon_{r}\left(\omega \right)=n^{2}$ where *n* is refractive index. In three dimensions, the Maxwell equations have six components: *E_x_*, *E_y_*, *E_z_* and *H_x_*, *H_y_*, and *H_z_*. If we assume that the structure is infinite, for example, in the *z* plane and that the fields are independent of *z*, in particular, we obtain [Equation (3)]:(3)$$\varepsilon_{r}\left(\omega ,x,y,z\right)=\varepsilon_{r}\left(\omega ,x,y,\right)\;\frac{\partial \vec{H}}{\partial t}=\frac{\partial \vec{E}}{\partial z}=0$$

The FDTD method uses finite differences as approximations to both spatial and temporal derivatives that appear in Maxwell’s equations (in particular, Ampère’s and Faraday’s laws). To move from an analytical solution to a numerical one, we consider the Taylor series expansion of the function *f(x)* expanded with respect to the point $x_{0}$ with a shift of ±*δ* = 2. Next, we combine the expansion into a Taylor series with the value +*δ* and −*δ*, divide by the error *δ*, and create a finite difference scheme [Equation (4)]:(4)$${\frac{\partial f}{\partial x}|}_{x=x_{0}}\approx \frac{f\left(x_{0}+\frac{\delta }{2}\right)-f\left(x_{0}-\frac{\delta }{2}\right)}{\delta }+O\left(\delta^{2}\right)$$

If *δ* is small enough, a reasonable approximation to the derivative can be obtained by simply neglecting all subsequent terms in the series [Equation (5)]:(5)$${\frac{\partial f}{\;\partial x}|}_{x=x_{0}}\approx \frac{f\left(x_{0}+\frac{\delta }{2}\right)-f\left(x_{0}-\frac{\delta }{2}\right)}{\delta }$$

The central difference provides an approximation of the derivative of the function at the point *x*_0_, to be precise, the function is chosen at the neighboring points $x_{0}$ + δ/2 and $x_{0}$ − δ/2 to approximate the vector function in Maxwell’s equations. Since the smallest power of the ignored *δ* is of the second order, this means that the central difference is of the second-order precision. Let us move on to the main governing equations when constructing the FDTD algorithm for a three-dimensional grid. These equations [Equations (6) and (7)] are:(6)$$-\sigma_{m}H-\mu \frac{\partial H}{\partial t}=\nabla \times E=\begin{array}{ccc} {\hat{𝕒}}_{x} & {\hat{𝕒}}_{y} & {\hat{𝕒}}_{z}\\ \frac{\partial }{\partial x} & \frac{\partial }{\partial y} & \frac{\partial }{\partial z}\\ E_{x} & E_{y} & E_{z} \end{array}$$
(7)$$\sigma E+\epsilon \frac{\partial H}{\partial t}=\nabla \times E=\begin{array}{ccc} {\hat{𝕒}}_{x} & {\hat{𝕒}}_{y} & {\hat{𝕒}}_{z}\\ \frac{\partial }{\partial x} & \frac{\partial }{\partial y} & \frac{\partial }{\partial z}\\ H_{x} & H_{y} & H_{z} \end{array}$$
where, *ε*–permittivity, *µ*–magnetic permeability, σ- electrical conductivity, and $\sigma_{m}$–magnetic conductivity. The three-dimensional mesh for modeling consists of six nodes [Equations (8)–(13)], which can be conventionally denoted by the indices *m*, *n*, and *p*.
(8)$$H_{x}(x;\;y;\;z;\;t)=H_{x}(m\Delta_{x};n\Delta_{y};p\Delta_{z};q\Delta_{t})=H_{x}^{q}[m;\;n;\;p],$$
(9)$$H_{y}(x;\;y;\;z;\;t)=H_{y}(m\Delta_{x};n\Delta_{y};p\Delta_{z};q\Delta_{t})=H_{y}^{q}[m;\;n;\;p],$$
(10)$$H_{z}(x;\;y;\;z;\;t)=H_{z}(m\Delta_{x};n\Delta_{y};p\Delta_{z};q\Delta_{t})=H_{z}^{q}[m;\;n;\;p],$$
(11)$$E_{x}(x;\;y;\;z;\;t)=E_{x}(m\Delta_{x};n\Delta_{y};p\Delta_{z};q\Delta_{t})=E_{x}^{q}[m;\;n;\;p],$$
(12)$$E_{y}(x;\;y;\;z;\;t)=E_{y}(m\Delta_{x};n\Delta_{y};p\Delta_{z};q\Delta_{t})=E_{y}^{q}[m;\;n;\;p],$$
(13)$$E_{z}(x;\;y;\;z;\;t)=E_{z}(m\Delta_{x};n\Delta_{y};p\Delta_{z};q\Delta_{t})=E_{z}^{q}[m;\;n;\;p].$$

Parts of a 3D cell are shown in Figure 1. This type of image is called a Yee cell. This cell consists of electric field nodes located along the edges of the cube and magnetic field nodes located on the faces.

With the arrangement of nodes shown in Figure 1, the components of Equations (6) and (7) can be expressed at the appropriate evaluation in Equations (14)–(19):(14)$$-\sigma_{m}H_{x}-\mu \frac{\partial H_{x}}{\partial t}={\frac{\partial E_{z}}{\partial y}-\frac{\partial E_{y}}{\partial z}|}_{x=m\Delta_{x},\;y=\left(n+\frac{1}{2}\right)\Delta_{y},z=\left(p+\frac{1}{2}\right)\Delta_{z},t=q\Delta_{t}},$$
(15)$$-\sigma_{m}H_{y}-\mu \frac{\partial H_{y}}{\partial t}={\frac{\partial E_{x}}{\partial z}-\frac{\partial E_{z}}{\partial x}|}_{x=(m+1/2)\Delta_{x},\;y=\left(n+\frac{1}{2}\right)\Delta_{y},z=\left(p+\frac{1}{2}\right)\Delta_{z},t=q\Delta_{t}\;},$$
(16)$$-\sigma_{m}H_{z}-\mu \frac{\partial H_{z}}{\partial t}={\frac{\partial E_{y}}{\partial x}-\frac{\partial E_{x}}{\partial y}|}_{x=(m+1/2)\Delta_{x},\;y=\left(n+\frac{1}{2}\right)\Delta_{y},z=\left(p+\frac{1}{2}\right)\Delta_{z},t=q\Delta_{t}},$$
(17)$$\sigma E_{x}+\epsilon \frac{\partial E_{x}}{\partial t}={\frac{\partial H_{z}}{\partial y}-\frac{\partial H_{y}}{\partial z}|}_{x=(m+1/2)\Delta_{x},\;y=n\Delta_{y},z=p\Delta_{z},t=\left(q+\frac{1}{2}\right)\Delta_{t}\;},$$
(18)$$\sigma E_{y}+\epsilon \frac{\partial E_{y}}{\partial t}={\frac{\partial H_{x}}{\partial z}-\frac{\partial H_{y}}{\partial x}|}_{x=m\Delta_{x},\;y=\left(n+\frac{1}{2}\right)\Delta_{y},z=p\Delta_{z},t=\left(q+\frac{1}{2}\right)\Delta_{t}\;},$$
(19)$$\sigma E_{z}+\epsilon \frac{\partial E_{z}}{\partial t}={\frac{\partial H_{y}}{\partial x}-\frac{\partial H_{x}}{\partial y}|}_{x=m\Delta_{x},\;y=n\Delta_{y},z=\left(p+\frac{1}{2}\right)\Delta_{z},t=\left(q+\frac{1}{2}\right)\Delta_{t}\;}$$

These equations ignore instantaneous losses and currents. The time derivative of each electric field component is always determined by the spatial derivative of the two magnetic field components and vice versa. In addition, the components of one field are associated with two orthogonal distributions of the components of the other field. As has been completed previously, the loss term can be approximated by the average of the field in two-times steps. Update equations for 3D meshes can be written by simply checking the underlying equations in continuous representation. Update equations in the final form [Equations (20)–(25)] are as follows:(20)$$H_{x}^{q+\frac{1}{2}}\left[m,n+\frac{1}{2}p,+\frac{1}{2}\right]=\frac{1-\frac{\sigma_{m}\Delta_{t}}{2\mu }}{1+\frac{\sigma_{m}\Delta_{t}}{2\mu }}H_{x}^{q-\frac{1}{2}}\left[m,n+\frac{1}{2},p+\frac{1}{2}\right]+\frac{1}{1+\frac{\sigma_{m}\Delta_{t}}{2\mu }}(\frac{\Delta_{t}}{\mu \Delta_{z}}\{E_{y}^{q}\left[m,n+\frac{1}{2},p+1\right]-E_{y}^{q}\left[m,n+\frac{1}{2},p\right]\}-\;-\frac{\Delta_{t}}{\mu \Delta_{z}}\left\{E_{z}^{q}\left[m,n+1,p+\frac{1}{2}\right]-E_{z}^{q}\left[m,n,p+\frac{1}{2}\right]\right\},$$
(21)$$H_{y}^{q+\frac{1}{2}}\left[m+\frac{1}{2},n,p+\frac{1}{2}\right]=\frac{1-\frac{\sigma_{m}\Delta_{t}}{2\mu }}{1+\frac{\sigma_{m}\Delta_{t}}{2\mu }}H_{y}^{q-\frac{1}{2}}\left[m+\frac{1}{2}n,p+\frac{1}{2}\right]+\frac{1}{1+\frac{\sigma_{m}\Delta_{t}}{2\mu }}(\frac{\Delta_{t}}{\mu \Delta_{z}}\{E_{z}^{q}\left[m+1,np+\frac{1}{2}\right]-E_{z}^{q}\left[m,n,p+\frac{1}{2}\right]\}-\;-\frac{\Delta_{t}}{\mu \Delta_{z}}\left\{E_{x}^{q}\left[m++\frac{1}{2},n,p+1\right]-E_{x}^{q}\left[m+\frac{1}{2},n,p\right]\right\},$$
(22)$$H_{z}^{q+\frac{1}{2}}\left[m+\frac{1}{2},n+\frac{1}{2},p\right]=\frac{1-\frac{\sigma_{m}\Delta_{t}}{2\mu }}{1+\frac{\sigma_{m}\Delta_{t}}{2\mu }}H_{z}^{q-\frac{1}{2}}\left[m+\frac{1}{2}n+\frac{1}{2},p\right]+\frac{1}{1+\frac{\sigma_{m}\Delta_{t}}{2\epsilon }}(\frac{\Delta_{t}}{\mu \Delta_{z}}\{E_{x}^{q}\left[m+\frac{1}{2},n+1,p\right]-E_{x}^{q}\left[m+\frac{1}{2},n,p\right]\}-\;-\frac{\Delta_{t}}{\epsilon \Delta_{z}}\{E_{y}^{q}\left[m++1,n+\frac{1}{2},p\right]-E_{y}^{q}\left[m,n+\frac{1}{2},p\right],$$
(23)$$E_{x}^{q+1}\left[m+\frac{1}{2},n,p\right]=\frac{1-\frac{\sigma \Delta_{t}}{2\epsilon }}{1+\frac{\sigma \Delta_{t}}{2\epsilon }}E_{x}^{q}\left[m+\frac{1}{2},n,p\right]+\frac{1}{1+\frac{\sigma \Delta_{t}}{2\epsilon }}(\frac{\Delta_{t}}{\epsilon \Delta_{z}}\{H_{z}^{q+\frac{1}{2}}\left[m+\frac{1}{2},n+\frac{1}{2},p\right]-H_{z}^{q+\frac{1}{2}}\left[m+\frac{1}{2},n-\frac{1}{2},p\right]\}-\;-\frac{\Delta_{t}}{\epsilon \Delta_{z}}\left\{H_{y}^{q+\frac{1}{2}}\left[m++\frac{1}{2},n,p+\frac{1}{2}\right]-H_{y}^{q+\frac{1}{2}}\left[m+\frac{1}{2},n,p-\frac{1}{2}\right]\right\},$$
(24)$$E_{y}^{q+1}\left[m,n+\frac{1}{2},p\right]=\frac{1-\frac{\sigma \Delta_{t}}{2\epsilon }}{1+\frac{\sigma \Delta_{t}}{2\epsilon }}E_{y}^{q}\left[m,n+\frac{1}{2}\right]+\frac{1}{1+\frac{\sigma \Delta_{t}}{2\epsilon }}(\frac{\Delta_{t}}{\epsilon \Delta_{z}}\{H_{x}^{q+\frac{1}{2}}\left[m,n+\frac{1}{2},p+\frac{1}{2}\right]-H_{x}^{q+\frac{1}{2}}\left[m,n+\frac{1}{2},p-\frac{1}{2}\right]\}-\;-\frac{\Delta_{t}}{\epsilon \Delta_{z}}\left\{H_{y}^{q+\frac{1}{2}}\left[m++\frac{1}{2},n+\frac{1}{2},p\right]-H_{z}^{q+\frac{1}{2}}\left[m-\frac{1}{2},n+\frac{1}{2},p\right]\right\}),$$
(25)$$E_{z}^{q+1}\left[m,n,p+\frac{1}{2}\right]=\frac{1-\frac{\sigma \Delta_{t}}{2\epsilon }}{1+\frac{\sigma \Delta_{t}}{2\epsilon }}E_{z}^{q}\left[m,n,p+\frac{1}{2}\right]+\frac{1}{1+\frac{\sigma \Delta_{t}}{2\epsilon }}(\frac{\Delta_{t}}{\epsilon \Delta_{z}}\{H_{y}^{q+\frac{1}{2}}\left[m+\frac{1}{2},n,p+\frac{1}{2}\right]-H_{y}^{q+\frac{1}{2}}\left[m-\frac{1}{2},n,p+\frac{1}{2}\right]\}-\;-\frac{\Delta_{t}}{\epsilon \Delta_{z}}\left\{H_{x}^{q+\frac{1}{2}}\left[m,n++\frac{1}{2},p+1/2\right]-H_{x}^{q+\frac{1}{2}}\left[m,n-\frac{1}{2},p+\frac{1}{2}\right]\right\}),$$

The coefficients in the update equations are assumed to be constant over time, but as in the case of our calculations, they can be functions of the position. In accordance with the accepted notation, and assuming a uniform grid (wherein, the grid in software simulation can be of different accuracy), in which ∆*x* = ∆*y* = ∆*z* = *δ*, the magnetic field update coefficients can be expressed as [Equations (26)–(31)]:(26)$$C_{hxh}\left(m,n+\frac{1}{2},p+\frac{1}{2}\right)={\frac{1-\frac{\sigma_{m}\Delta_{t}}{2\mu }}{1+\frac{\sigma_{m}\Delta_{t}}{2\mu }}|}_{m\delta ,\left(n+\frac{1}{2}\right)\delta ,\left(p+\frac{1}{2}\right)\delta },$$
(27)$$C_{hxe}\left(m,n+\frac{1}{2},p+\frac{1}{2}\right)={\frac{1-\frac{\sigma_{m}\Delta_{t}}{2\mu }}{1+\frac{\sigma_{m}\Delta_{t}}{2\mu }}\frac{\Delta_{t}}{\mu \delta }|}_{m\delta ,\left(n+\frac{1}{2}\right)\delta ,\left(p+\frac{1}{2}\right)\delta },$$
(28)$$C_{hyh}\left(m+\frac{1}{2},n,p+\frac{1}{2}\right)={\frac{1-\frac{\sigma_{m}\Delta_{t}}{2\mu }}{1+\frac{\sigma_{m}\Delta_{t}}{2\mu }}|}_{\left(m+\frac{1}{2}\right)\delta ,n\delta ,\left(p+\frac{1}{2}\right)\delta },$$
(29)$$C_{hye}\left(m+\frac{1}{2},n,p+\frac{1}{2}\right)={\frac{1}{1+\frac{\sigma_{m}\Delta_{t}}{2\mu }}\frac{\Delta_{t}}{\mu \delta }|}_{\left(m+\frac{1}{2}\right)\delta ,n\delta \left(p+\frac{1}{2}\right)\delta },$$
(30)$$C_{hzh}\left(m+\frac{1}{2},n+\frac{1}{2},p\right)={\frac{1-\frac{\sigma_{m}\Delta_{t}}{2\mu }}{1+\frac{\sigma_{m}\Delta_{t}}{2\mu }}|}_{\left(m+\frac{1}{2}\right)\delta ,\left(n+\frac{1}{2}\right)\delta ,p\delta },$$
(31)$$\;\;\;\;\;\;\;\;\;\;\;\;C_{hze}\left(m+\frac{1}{2},n+\frac{1}{2},p\right)={\frac{1}{1+\frac{\sigma_{m}\Delta_{t}}{2\mu }}\frac{\Delta_{t}}{\mu \delta }|}_{\left(m+\frac{1}{2}\right)\delta ,\left(n+\frac{1}{2}\right)\delta ,p\delta },$$

For the electric-field update equations, the coefficients are in the following form [Equations (32)–(37)]: (32)$$C_{exe}\left(m+\frac{1}{2},n,p\right)={\frac{1-\frac{\sigma \Delta_{t}}{2\epsilon }}{1+\frac{\sigma \Delta_{t}}{2\epsilon }}|}_{\left(m+\frac{1}{2}\right)\delta ,n\delta ,p\delta },$$
(33)$$C_{exh}\left(m+\frac{1}{2},n,p\right)={\frac{1}{1+\frac{\sigma \Delta_{t}}{2\epsilon }}\frac{\Delta_{t}}{\epsilon \delta }|}_{\left(m+\frac{1}{2}\right)\delta ,n\delta ,p\delta },$$
(34)$$C_{eye}\left(m,n+\frac{1}{2},p\right)={\frac{1-\frac{\sigma \Delta_{t}}{2\epsilon }}{1+\frac{\sigma \Delta_{t}}{2\epsilon }}|}_{m\delta ,\left(n+\frac{1}{2}\right)\delta ,p\delta },$$
(35)$$\;C_{eyh}\left(m,n+\frac{1}{2},p\right)={\frac{1}{1+\frac{\sigma \Delta_{t}}{2\epsilon }}\frac{\Delta_{t}}{\epsilon \delta }|}_{m\delta ,\left(n+\frac{1}{2}\right)\delta ,p\delta },$$
(36)$$\;C_{eze}\left(m,n,p+\frac{1}{2}\right)={\frac{1-\frac{\sigma \Delta_{t}}{2\epsilon }}{1+\frac{\sigma \Delta_{t}}{2\epsilon }}|}_{m\delta ,n\delta ,\left(p+\frac{1}{2}\right)\delta },$$
(37)$$C_{ezh}\left(m,n,p+\frac{1}{2}\right)={\frac{1}{1+\frac{\sigma \Delta_{t}}{2\epsilon }}\frac{\Delta_{t}}{\epsilon \delta }|}_{m\delta ,n\delta ,\left(p+\frac{1}{2}\right)\delta }.$$

These coefficients can be solved with the Courant number, which is related to the convergence condition of this equation. For a uniform grid in three dimensions, the Courant limit must be equal to 1/√3. We will not touch upon a rigorous derivation of this limit. Let us use a simple empirical explanation. To ensure stability, we need to make sure that the distance traveled in the real continuous world in three-time steps is less than the distance over which the grid can transmit information. The grid must have time to calculate the propagation of the same perturbation no slower than if this perturbation took place in real time. In our case, we use the most accurate standard grid (not counting the additional refinement grid).

### 2.2. Raman Scattering and Enhancement Factor

The Raman scattering effect is the result of inelastic scattering between a photon and the vibrational modes of a molecule. The power of the scattered Raman signal can be described by the expression [Equation (38)]: (38)$$P_{s}\left(v_{s}\right)=N\sigma_{RS}I\left(\nu_{L}\right),$$
where N denotes the number of active scatterings within the propagated excitation, $\sigma_{RS}$ denotes the scattering cross section, $I\left(\nu_{L}\right)$ is the intensity of the incident beam at the frequency $\nu_{L}$, and $v_{s}$ is the frequency of the scattered Raman signal. This expression takes into account only the Stokes shift, in which case the Raman signal will be less than the frequency of the incident light [22]. However, in Raman spectroscopy, the scattering intensity can be put into a linear dependence on the intensity of the incident field,$\;E_{0}^{2}$. The Stokes shift occurs when an incident photon interacts with a molecule in its ground vibrational state. Near the surface of scattering metal nanostructures, this process is accompanied by signal amplification and is known as SERS [23]. Since the magnitude of the field intensity on these surfaces is significantly increased, the intensity of Raman scattering can be related to the absolute value of the square $E_{out}$ on the surface of the NPs. Let us give, as an example, the equation for a metal sphere [Equation (39)] [24]:(39)$$\left[{\left|E_{out}\right|}^{2}=E_{0}^{2}[{\left|1-g\right|}^{2}+3cos^{2}\theta \left(2Re\left(g\right)+{\left|g\right|}^{2}\right)\right],$$
where g denotes a value that depends on the dielectric constant of the medium and the metal NP and θ is the angle between the incidence field vector and the vector directed to the position of the molecule on the surface. Peak amplification occurs when θ is 0° or 180°, which corresponds to the position on the axis of light propagation. In cases where g is large, the maximum gain approaches [Equation (40)] [24]:(40)$${\left|E_{out}\right|}^{2}=4\;E_{0}^{2}{\left|g\right|}^{2}.\;$$

We take into account the contribution of the applied field to Raman scattering which, as noted above, induces a vibrational dipole in the molecule on the surface. Then, this dipole radiates, and in the approximation there is a probability that the emitted light is, as in theory, shifted by the vibrational frequency of the molecule. The first order approximation is to use an expression similar to Equations (14)–(19), except that it is evaluated at the Raman–Stokes bias frequency, so the following expression can be written in approximation [Equation (41)] [24] as follows:(41)$$EF=\frac{{\left|E_{out}\right|}^{2}{\left|\;E_{out}^{\prime }\right|}^{2}}{|E_{0}{|}^{4}}=4{\left|g\right|}^{2}{\left|g^{\prime }\right|}^{2}.$$

This expression is defined as the theoretical gain (*EF*) of the SERS. In the literature, this expression is called the fourth power of field enhancement on the surface of NPs ${\left|E\right|}^{4}$. More details on the use of the coefficient ${\left|E\right|}^{4}$ can be found, for example, in these works [24,25].

This section shows the main essence of the use of *EF* SERS related directly to the interpretation of experimental data. In our work, however, a slightly different theoretical model of *EF* is used, as described in Section 2.3, not including the presence of the analyte and depending largely on the strength of the electric field and the incident wave.

### 2.3. Simulation Process

Modeling was performed using the Lumerical FDTD Solutions software package [v.8.19.1584, Lumerical Inc. (Vancouver, BC, Canada)]. The simulation was carried out for rod-shaped Au NPs (NRs). The NRs were varied radii of 5, 10, 15, and 20 nm. NRs’ lengths were taken as 70 and 80 nm. Moreover, NRs of the same dimensions were modeled using an encapsulating SiO_2_ core–shell shell with shell thicknesses of 2, 3, 5, 10, 15, and 20 nm. Schematic illustrations of NRs FDTD simulations are illustrated in Figure 2.

The stages of the simulation process were performed as follows:

(1) The counting area, grid resolution, and boundary conditions were set. For the computational domain, a rectangular grid was used from the basic Yee algorithm in the Cartesian coordinate system. The main modeling quantities (material properties and object geometry, electric, and magnetic fields) were calculated separately at each grid point. The size of the computational area along the axes varied within different limits based on the size of the objects. To maintain accuracy, the meshing algorithm generated a smaller mesh with a high index (to maintain a constant number of mesh points per wavelength). The minimum grid spacing was set to 0.25 nm. Then, an additional refinement mesh was installed for the simulation. The size of the computational region of the additional grid was set by the grid step: *dx, dy,* and *dz* = 2.5 nm. We have chosen standard absorbing perfectly matched layer (PML) and boundary conditions designed to absorb incident light with minimal reflections. Their parameters were as follows:−Layers (for PML area discretization purposes) = 8.−KAPPA, SIGMA, and ALPHA (absorptive properties of PML regions kappa, sigma, and alpha are estimated inside PML regions using polynomial functions) kappa = 2, sigma = 1, and alpha = 0.−Polynom (defines the order of the polynomial used to evaluate kappa and sigma) = 0.−Alpha-polynomial (defines the order of the polynomial used to evaluate the alpha channel) = 1.−Minimum and maximum layers (these provide an acceptable range of values for the number of PML layers). Minimum layers = 8 and maximum = 64.−Physical parameters of the simulation: travel time of a plane-polarized wave through the working area 1000 fs and temperature T = 300 °K.

(2) A body with specified optical and geometrical parameters was placed inside the counting region. Next, the optical and geometric parameters of the samples were set. We used materials from a Lumerical database (Au, SiO_2_) and changed their parameters (size, shape, and geometry) for modeling. The values of such a parameter as the real and imaginary parts of the permittivity, which depends on the radiation frequency, were taken into account. In our case, a plane wave equal to λ = 532, 632.8, and 785 nm. We also used theoretical ε (Au) values for modeling, taken from the Lumerical database (Table 1).

(3) At the next step, the radiation source parameters were set for three wavelengths. In our study, a total scattered field source (TFSF) is used, which is often suitable for studying scattering by small particles illuminated by a plane wave. The TFSF source divides the computational domain into two separate domains: (a) the total field domain includes the sum of the incident field wave plus the scattered field, and (b) the scattered field domain includes only the scattered field. The TFSF source is an extended source. It is important to note that the physical field is a total field, and the division into the incident and scattered fields requires careful interpretation. For NPs in a homogeneous medium, the incident field is a *p*-polarized plane wave. We obtained the magnitude of the electric tension in the maximum values. We also calculated the array of electric field values in the region of plasmon generation defined by us as the integral sum $\overset{k}{\underset{p}{\sum }}E$ attenuation point of the propagated field. It should be noted that the position of the source of a plane *p*-polarized wave is important for calculating the maximum electric field value, especially under conditions of single measurements, but this is not decisive for the overall distribution of the dependence in a rather large sample. We have installed the radiation source close to the NR at 3/2 of its length.

(4) To provide final information, the monitor plane was set perpendicular to the *xz*-plane, which gave us the final information about the value of the electric field as a function of position in space, in the form of a 2D slice. The use of field monitors in the frequency domain allowed us to collect a field profile in this region and provide simulation results in some spatial domains to the FDTD solver. The wave was polarized along the *z* axis, i.e., its direction corresponded to the normal vector to the monitor surface on the XZ axis.

(5) As the last step, the calculated values of the electric field were converted using the Lumerical program code (scripts) into the intensity values of the Raman and SERS according to the theoretical effective ${\left|E/E_{0}\right|}^{4}$ or enhancement factor (*EF*) for SERS calculated in the *xz*-plane. In order to do this, we use TFSF sources and models in this area of the structure to find the maximum values of *EF*. Even though a SERS calculation is performed mostly for rough surfaces or dimers on which it is calculated at hot spots between gaps, we modeled it for sheathed NRs. Comparing such NRs with NRs without shells, we can check the LSPR presence in NPs by the presence of a dielectric shell. The theoretical gain has been calculated for NRs with/without shells. This parameter, in contrast to experimental measurements, can be calculated without conditions for the presence of the analyte.

Let us take into account the absence as such of a strict dependence between *E* as the magnitude of the electric field strength of localized surface plasmons, which for a TFSF source determines mainly the magnitude of the total plus scattered field, to the SERS scattering intensity and *EF* coefficient, which depend on the resonant absorption frequency of various gold NPs. It is possible that the integral sum of the electric fields distributed near the surface of the NRs will approximate this dependence. The integral sum of field $\overset{k}{\underset{p}{\sum }}E\;$ is defined in relative terms, as it depends on the NR radius. The SERS and *EF* values themselves are equivalent in our theoretical modeling, i.e., one value of SERS will correspond to only one value of *EF*. However, in experimental studies, there are inconsistencies between these values due to the lack of strict definitions [26].

## 3. Simulation Results and Discussion

### 3.1. NR without a SiO_2_ Shell

As a result of the first part of the simulation, the values of the local maximum of the electric field were obtained as a function of SERS and *EF* for Au NRs with a thickness of 5, 10, 15, and 20 nm. Depending on the NR radius, three radiation wavelengths were used: 532, 632.8, and 785 nm. NRs of two lengths (70 and 80 nm) were applied in experiments. This is due to insignificant differences in the studied values when varying this parameter, since all parameters were calculated in the plane parallel to the NR length (*xz*). In this line of modeling, we did not calculate the integral sum of the fields, which is mostly comparative in nature with a large sample. The data are summarized in Table 2.

It can be seen that the NRs give low values of electric field strength, mainly increasing with NR radius growth. A similar increase also occurs in the analysis of identical NRs with an increase in the radiation wavelength. The main results for NR with a 70 nm and 80 nm length for different excitation sources are shown in Figure 3.

The SERS intensity and *EF* appeared to increase with the NR radius, which was manifested significantly at a 785 excitation wavelength. That fact correlates with [27] data and depends on plasmon excitation maximum near 785 nm. 

### 3.2. NR with a SiO_2_ Shell

The second series of modeling experiments were carried out with the Au@SiO_2_ complex using a *xz* monitor plane. Below are the data obtained for Au NRs with radii of 5, 10, 15, and 20 nm and a SiO_2_ shell of various thicknesses (2, 3, 5, 10, 15, and 20 nm). The same parameters were studied for NRs without a shell, to which the study of the integral sum of fields was added. The effect of different shell thicknesses, as expected, introduced a nonlinear character into the quantities under study (Table 3).

When analyzing the results obtained from the electric field simulation for NRs with a SiO_2_ shell, it can be seen that the electric field values for three different excitation wavelengths (532, 632.8, and 785 nm) showed a similar distribution as the "bare" NRs. An increase occurred for relatively thin NRs (5 and 10 nm radii), smoothing out the differences at 632.8 and 785 nm for thick NRs (15 and 20 nm radii). For SERS and *EF*, the values were approximately equal for the same radii at 532 and 632.8 nm for thin NRs (5 and 10 nm radii) increasing in difference in favor of 532 with radius growth. Moreover, SERS/*EF* values appeared to become higher at the 785 nm excitation wavelength. Analyzing the influence of the SiO_2_ shell, we did not observe a clear monotonous character: electric field and SERS signal intensities fell at medium values of SiO_2_ thickness. First, in comparison with bare NRs, shelled NRs excited at wavelengths of 532 and 632.8 nm had lower electric field strengths for thin-coated NRs and higher values for thick NRs coated with a thick SiO_2_ shell. This can be explained by the fact that the thickness of the SiO_2_ shell in the case of NRs excited by laser radiation of the aforementioned wavelengths affects the refractive index by changing the extinction coefficient. This can explain the relatively low values of maximum electric field strength at an average shell thickness. For the 532 nm excitation wavelength, the largest values of the electric field almost always corresponded to the relatively largest values of SERS. No such correspondence was observed for the 632.8 and 785 nm excitation wavelengths. This can be explained by the fact that such sizes of sheathed gold rods can fall into absorption peaks in the long wavelength region and contribute to SERS, but they contribute less to the scattered field. For the NRs irradiated by the 785 nm laser emission, an increase in the thickness of the shell leads to an increase in the difference between the maximum values of the electric field and the relatively maximum SERS. This is consistent with the fact that such sizes of sheathed gold NRs fall into absorption peaks in the long wavelength region and contribute to SERS, but do not contribute to scattering. 

### 3.3. A Single (hollow) SiO_2_ Shell

The most interesting case was the simulation of the silica shell scattering without the NR (single, or hollow NR). The quantitative contribution was determined for the scattering of the silica shell in the overall pattern of changes for the simulated parameters of the electric field. The obtained results show that silica shells contribute to the scattering of the overall system at all excitation wavelengths used in our simulation. These results (Table 4) correlate with [23,28]. 

Based on the results of the paper, it was found that the silicon shell, although it contributes to scattering, cannot be used as an independent medium for a significant increase or decrease in the scattering of structures adsorbed on it. In addition, as the shell thickness increases, the scattering decreases, from which we can conclude that the overall efficiency of Au@SiO_2_ sensors decreases.

### 3.4. Calculation of the Integral Sum for a Single (Hollow) SiO_2_ and NR Au@SiO_2_

The maximum values of the electric field integral sums almost always correspond either to the maximum values of *E*, which is logically explained by the fact that one of the values taken will be included in the largest, or to the high values of SERS, which can be explained by the fact that SERS itself is nothing more than the sum of the peaks of the Raman signal shift.

The simulations of the electrical field integral sum distributed near the surface of the NRs and shell demonstrate strong *E* presence in the system. At 632.8 and 785 nm excitation wavelengths (Table 5), electrical field strength values are higher than at 532 nm (for both 70 and 80 nm NR lengths). For 5 and 10 nm inner shell radii, *E* decreases in Au@SiO_2,_ but in SiO_2_, *E* did not show any obvious correlations with NR or shell size parameters. 

## 4. Conclusions

As a result, a flexible simulation algorithm based on the FDTD method has been developed for the simulation of electric field parameters in each component of the Au@SiO_2_ system. It has been found that the SiO_2_ shell on Au NR, although it contributes to the total system scattering, demonstrates low values. Some cases linearly increased from the thickness of the SiO_2_ shell at the laser excitation wavelength of λ = 532 nm and λ = 632.6 nm, because for λ = 785 nm laser excitation there is an absorption peak. The highest value of the *E* = 6.29 V/m was revealed for nanorods without a shell, excited by a monochromatic wave λ = 785 nm using dimensional characteristics: length 80 nm and radius 20 nm. The parameter ${\left|E/E_{0}\right|}^{4}$ = 1.41 × 10^7^ was obtained for a rod with almost the same parameters, except for the length (l = 70 nm). On the hollow SiO_2_ shells, the highest *E* value (*E* = 2.81 V/m) was obtained with the simulation parameters of length 70 nm, inner radius 20 nm, and shell thickness 20 nm, upon excitation with a wavelength of λ = 785 nm. The highest value for the ${\left|E/E_{0}\right|}^{4}$ = 1.8 × 10^5^ was obtained with the simulation parameters of length 70 nm, inner radius 5 nm, and shell thickness 15 nm, with excitation at a wavelength of λ = 785 nm. For the SiO_2_ coated rod, the highest maximum values of the electric field strength (*E* = 7.34 V/m), as well as the ${\left|E/E_{0}\right|}^{4}$ = 3.15 × 10^7^), when excited by laser radiation λ = 785 nm, with dimensional parameters were length 70 nm, rod width 20 nm, and shell thickness 20 nm. As we can see, this simulation object had the highest values of all. As a result, the electric field parameters of Au NR, Au@SiO_2_, and a single (hollow) SiO_2_ shell were calculated and estimated. The results of the work can be used in methods for the synthesis of core–shell NPs.

## Figures and Tables

**Figure 1 nanomaterials-12-04011-f001:**
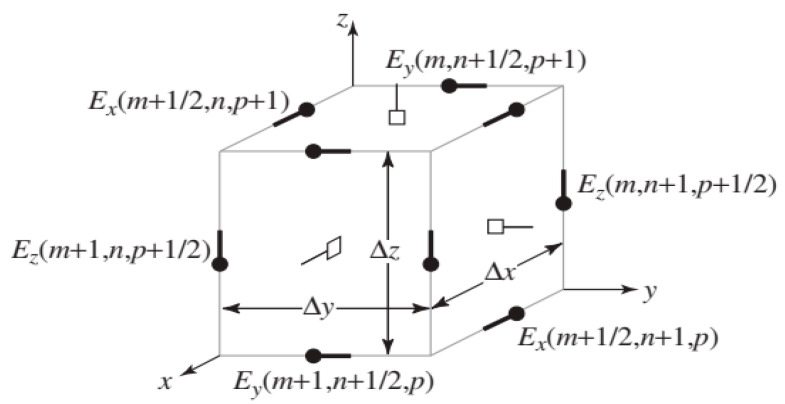
Nodes in a 3D FDTD mesh as a Yee cell. In this image, not all nodes have the same indexes. As shown here, the cube will consist of four *E_x_* nodes, four *E_y_* nodes, and four *E_z_* nodes, i.e., the electric fields are located along the edges of the cube. The magnetic fields are on the faces of the cube and hence there will be two *H_x_* nodes, two *H_y_* nodes, and two *H_z_* nodes.

**Figure 2 nanomaterials-12-04011-f002:**
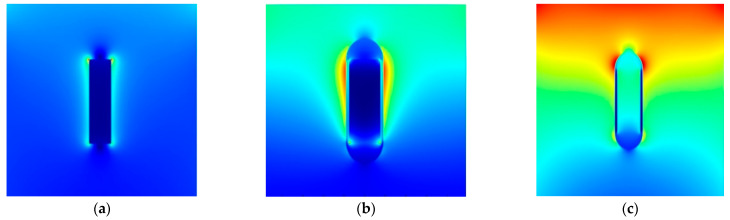
Schematic illustrations of NR FDTD simulations. The examples show theoretical simulations for NRs without a SiO_2_ shell [(**a**)–cylinder (Au) height-70 nm, radius-10 nm, excitation wavelength-532 nm)], for NRs with a SiO_2_ shell [(**b**)–cylinder (Au) height-70 nm, radius-15 nm, the thickness of SiO_2_-3 nm, excitation wavelength-532 nm)], and a single (hollow) SiO_2_ shell [(**c**)–height-70 nm, the thickness of SiO_2_-3 nm, radius-15 nm, and excitation wavelength-532 nm].

**Figure 3 nanomaterials-12-04011-f003:**
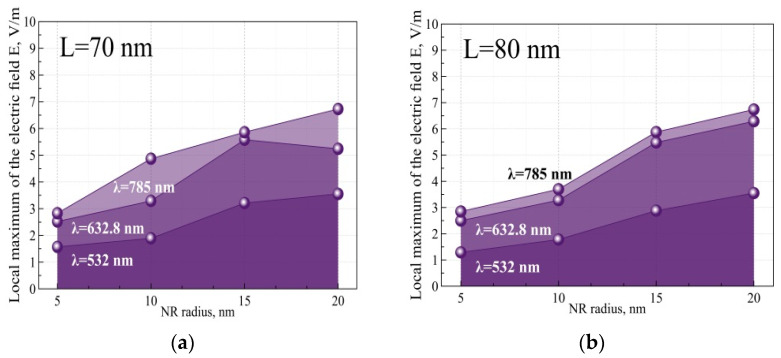
Maximum values of electric field tension were calculated for NRs with 70 (**a**) and 80 nm (**b**) lengths and an NR radius of 5, 10, 15, and 20 nm. Three excitation sources were used: 532 nm, 632.8 nm, and 785 nm.

**Table 1 nanomaterials-12-04011-t001:** Electric field simulation parameters for NR without SiO_2_.

Excitation Wavelength λ, nm	Re (ε)	Im (ε)
532	−4.29	1.64
632.8	−10.8	0.795
785	−21.64	0.74

**Table 2 nanomaterials-12-04011-t002:** Electric field simulation parameters for NRs without SiO_2_.

NR Length, nm	NR Radius	Excitation Wavelength, nm								
		532	632.8	785	532	632.8	785	532	632.8	785
		Local Maximum of the Electric Field *E*, V/m			SERS Signal Intensity, a,u			Enhancement Coefficient $10^{4}\;{\left|E/E_{0}\right|}^{4}$		
70	5	1.57	2.52	2.83	110	90	1040	1.22	0.8	110
	10	1.89	3.29	4.87	140	133	2620	2	1.8	700
	15	3.21	5.58	5.86	462	500	3330	22	24	1100
	20	3.55	5.24	6.73	560	340	3770	29.5	12	1410
80	5	1.29	2.5	2.85	109	105	1020	1.12	1.1	103
	10	1.78	3.28	3.7	120	127	1300	1.43	1.6	165
	15	2.88	5.48	5.88	455	460	3150	21	21.4	1000
	20	3.55	6.29	6.74	520	530	3600	26.6	27.5	1320

**Table 3 nanomaterials-12-04011-t003:** Electric field simulation parameters for NRs with SiO_2_.

NR Length, nm	NR Radius nm	Shell Thickness nm	532	632.8	785	532	632.8	785	532	632.8	785
			Local Maximum of the Electric Field *E*, V/m			SERS Signal Intensity, a,u			Enhancement Coefficient $10^{4}\;{\left|E/E_{0}\right|}^{4}$		
70	5	2	1.53	2.31	2.80	112	90	491	1.27	0.8	24.2
		3	1.28	2.29	2.74	104	88	462	1.1	0.74	21.9
		5	1.25	2.26	2.67	100	95	458	1	0.9	21.1
		10	1.18	2.25	2.71	99	97	435	0.99	0.93	19.1
		15	1.20	2.30	2.74	95	100	433	0.9	0.1	19
		20	1.39	2.27	2.78	81	74	329	0.7	0.55	11
	10	2	1.68	2.83	3.44	190	142	630	3.6	2.01	39
		3	16.3	2.68	3.16	162	130	1310	2.7	1.63	172
		5	1.58	2.64	3.07	120	137	1350	1.41	1.85	182
		10	1.39	2.58	2.97	150	116	1260	2.21	1.34	158
		15	1.45	2.58	2.93	169	104	710	2.8	1.1	50
		20	1.57	2.52	2.88	165	123	730	2.71	1.51	53
	15	2	3.23	3.83	5.92	700	520	2410	48.5	27	600
		3	3.66	3.70	5.43	620	379	2290	39	14.4	520
		5	3.72	4.38	5.06	710	610	3790	49.1	37.2	1490
		10	3.91	4.62	5.17	1050	910	4800	110	83	2340
		15	3.78	4.93	5.54	1100	1200	5400	120	145	2800
		20	4.07	6.14	5.68	1320	1360	4400	173	185	1950
	20	2	3.47	6.34	6.68	840	550	2430	70	29	600
		3	3.35	6.10	6.38	620	493	2600	38	24.4	700
		5	3.64	5.96	5.68	900	905	3810	80	83	1500
		10	3.84	6.48	5.79	1200	1100	4800	155	114	2400
		15	3.91	6.92	6.28	1500	1500	5400	220	223	2800
		20	3.97	7.34	6.55	1550	1700	6000	243	283	3150
80	5	2	1.45	2.49	2.70	110	100	990	1.14	1	930
		3	1.39	2.35	2.66	102	95	910	10.4	0.9	850
		5	1.39	2.32	2.58	100	91	920	9.6	8.4	860
		10	1.32	2.29	2.58	99	92	910	9.5	0.89	850
		15	1.27	2.22	2.62	99	109	910	9.5	11.2	850
		20	1.31	2.19	2.63	80	80	700	0.6	6.6	435
	10	2	1.51	2.84	3.45	190	161	600	3.5	2.6	35
		3	1.53	2.79	3.37	164	150	670	2.7	2.16	42.1
		5	1.41	2.68	3.26	140	114	690	1.9	1.3	45.2
		10	1.43	2.58	3.17	142	114	890	2	1.3	77
		15	1.45	2.56	3.11	160	119	400	2.5	1.35	16.1
		20	1.64	2.35	3.06	180	173	380	3.15	3	14.5
	15	2	3.19	5.45	6.00	500	600	2520	25.2	33.2	650
		3	3.50	5.39	5.70	600	478	2500	35	23	620
		5	3.68	5.16	5.12	570	421	3810	29	18.2	1470
		10	3.89	5.50	5.10	700	432	4820	45	19	2350
		15	3.94	6.01	5.56	840	530	5400	70	27.3	2840
		20	4.02	5.87	5.71	990	530	3200	93	27.3	1040
	20	2	3.20	6.35	6.58	600	640	2820	33.1	40.5	800
		3	3.37	6.05	6.30	680	600	2610	42.4	33.1	700
		5	3.67	5.82	5.60	600	520	2100	34.2	27	422
		10	3.89	6.13	5.74	660	580	2800	41.7	31	795
		15	3.99	6.65	6.19	750	720	2820	60	52	805
		20	4.02	7.07	6.40	990	890	3600	95	76	1300

**Table 4 nanomaterials-12-04011-t004:** Electric field simulation parameters for a SiO_2_ shell.

Shell Length, nm	Inner Shell Radius, nm	Shell Thickness, nm	532	632.8	785	532	632.8	785	532	632.8	785
			Local Maximum of the Electric field *E*, V/m			SERS Signal Intensity, a,u			Enhancement Coefficient $10^{4}\;{\left|E/E_{0}\right|}^{4}$		
70	5	2	1.11	1.91	2.20	63	56.5	121	0.395	0.32	1.45
		3	1.16	1.94	2.26	70	62.1	130	0.43	0.38	1.7
		5	1.16	1.99	2.24	72	57.6	145	0.5	0.33	2.1
		10	1.10	2.06	2.57	80	78	385	0.63	0.58	15
		15	1.32	2.12	2.65	90	94	420	0.77	0.89	18
		20	1.29	2.04	2.78	51.4	71	268	0.27	0.45	6.9
	10	2	1.02	1.85	2.43	61	74.6	286	0.36	0.56	8.3
		3	1.03	1.87	2.46	62.5	71	292	0.39	0.51	8.7
		5	1.00	1.91	2.57	70	70	338	0.45	0.45	11.3
		10	1.02	2.03	2.62	73	90	370	0.53	0.8	13.6
		15	1.06	1.90	2.75	48.5	81	331	0.24	0.62	11
		20	1.22	1.90	2.79	57.5	85	400	0.33	0.71	16
	15	2	0.97	1.95	2.54	53.2	67.5	267	0.28	0.45	7.13
		3	0.94	1.85	2.61	60	78	312	0.36	0.59	9.9
		5	1.01	1.92	2.61	60.6	86	380	0.37	0.74	14.5
		10	1.18	1.99	2.62	56	58	130	0.31	0.34	1.65
		15	1.23	2.05	2.71	62.8	59.9	140	0.40	0.37	1.9
		20	1.18	2.14	2.77	61.6	70	164	0.38	0.44	2.69
	20	2	1.04	1.91	2.64	44.4	61	236	0.2	0.37	5.55
		3	1.10	1.89	2.61	49.8	61	230	0.25	0.37	5.3
		5	1.00	1.93	2.64	47	58.5	230	0.22	0.34	5.3
		10	1.10	1.98	2.71	54.2	56.5	140	0.29	0.32	1.95
		15	1.11	1.96	2.73	52.4	70	129	0.27	0.43	1.63
		20	1.10	1.97	2.81	57.5	67	159	0.33	0.42	2.5
80	5	2	1.11	1.99	2.24	63	56	124	0.39	0.32	1.53
		3	1.18	1.94	2.25	70	62	130	0.43	0.38	1.7
		5	1.18	1.99	2.32	71	57.5	140	0.49	0.33	1.91
		10	1.10	2.05	2.52	80	78	370	0.63	0.58	14
		15	1.31	2.11	2.63	89	93	410	0.76	0.85	16.2
		20	1.28	2.03	2.72	51.3	70	253	0.26	0.45	6.5
	10	2	1.01	1.84	2.46	60	74.5	280	0.36	0.55	8
		3	0.99	1.86	2.45	61.5	71.1	291	0.38	0.52	8.5
		5	1.00	1.88	2.50	62	61	330	0.38	0.45	11
		10	1.02	2.02	2.59	73	90	365	0.53	0.8	13.5
		15	1.05	1.89	2.69	48	80	333	0.24	0.6	11.3
		20	1.21	1.89	2.73	57.5	85	415	0.33	0.71	16.5
	15	2	0.96	1.94	2.59	52.4	67.5	270	0.27	0.45	7.2
		3	0.93	1.73	2.56	60	78	310	0.36	0.59	9.5
		5	1.01	1.9	2.58	60.5	86	370	0.36	0.74	13.7
		10	1.17	1.98	2.61	56.2	58	130	0.31	0.34	1.65
		15	1.22	2.035	2.7	63	61	140	0.4	0.37	1.9
		20	1.18	2.14	2.71	61	70	165	0.37	0.44	2.7
	20	2	1.03	1.9	2.63	45	62	239	0.2	0.38	5.5
		3	1.08	1.875	2.62	50	61.5	232	0.25	0.37	5.35
		5	0.99	1.92	2.64	47.1	59	233	0.22	0.34	5.36
		10	1.07	1.975	2.7	53.5	57	140	0.29	0.32	1.95
		15	1.01	1.96	2.73	52.5	70	130	0.28	0.43	1.65
		20	1.09	2.04	2.8	58	66	160	0.34	0.42	2.52

**Table 5 nanomaterials-12-04011-t005:** The integral sum of the electrical field is distributed near the surface of the NRs and SiO_2_ shells.

NR Length, nm	Inner Shell Radius, nm	Shell Thickness, nm	Au@SiO_2_			Single (Hollow) SiO_2_		
			532	632.8	785	532	632.8	785
			$\overset{k}{\underset{p}{\sum }}E\;relative\;units$					
	5	2	15.0	23.0	19.1	12.8	26.8	32.3
		3	15.3	24.2	19.3	10.8	26.5	31.2
		5	14.0	23.8	18.9	10.7	26.4	30.9
		10	13.0	24.3	20.4	10.4	25.7	30.8
		15	15.8	25.1	21.5	9.6	27.3	29.9
		20	14.3	23.4	22.2	10.3	25.5	30.2
	10	2	11.8	21.0	23.8	12.6	25.1	31.4
		3	11.1	21.4	23.9	12.1	24.1	29.5
		5	12.9	21.5	25.6	11.8	23.3	29.6
		10	11.7	22.7	27.0	11.6	22.4	28.2
		15	8.5	21.9	27.6	12.2	21.5	26.9
		20	10.2	21.8	28.0	12.3	22.0	26.5
	15	2	9.4	22.6	23.4	16.0	28.0	39.9
		3	8.4	22.2	23.2	17.5	28.3	39.8
		5	9.0	23.3	24.7	16.2	29.0	41.4
		10	8.3	23.0	25.6	18.1	29.2	41.6
		15	12.1	21.7	26.3	18.1	29.7	41.0
		20	9.5	22.7	28.6	20.5	31.4	39.4
	20	2	16.9	19.6	27.1	16.6	38.0	37.5
		3	17.1	19.5	27.0	18.4	33.3	38.3
		5	17.3	19.4	26.4	18.3	36.1	38.2
		10	17.4	19.9	27.2	19.8	36.9	37.4
		15	17.7	21.3	27.4	19.6	37.7	37.4
		20	17.9	21.6	28.4	20.2	39.6	38.7
80	5	2	20.6	21.2	21.2	14.2	28.1	19.2
		3	20.0	22.4	21.2	13.9	29.1	18.3
		4	19.2	22.1	21.1	13.2	28.9	17.7
		10	20.2	22.5	22.8	12.7	28.0	16.3
		15	20.8	23.2	24.0	11.3	28.1	17.1
		20	19.4	22.5	25.0	10.6	26.6	17.4
	10	2	20.2	19.4	28.0	15.6	28.6	21.7
		3	17.7	19.7	28.0	14.7	27.6	19.9
		5	22.2	20.6	27.9	13.5	27.4	19.4
		10	17.4	21.6	29.0	14.3	26.5	23.0
		15	17.0	20.1	30.2	14.6	27.8	20.7
		20	19.2	21.7	30.8	15.3	27.8	20.8
	15	2	15.3	19.3	27.9	18.5	30.6	33.1
		3	15.0	20.0	27.8	20.3	31.2	33.0
		5	15.9	20.0	29.4	22.1	32.0	33.2
		10	18.7	21.1	27.9	21.2	33.4	31.9
		15	18.5	21.6	29.0	22.2	33.4	34.1
		20	17.3	22.7	28.7	22.9	32.6	34.8
	20	2	15.8	19.5	31.4	19.6	33.4	37.5
		3	16.8	19.3	31.4	20.0	33.0	37.9
		5	15.1	20.9	30.9	20.5	34.1	37.8
		10	15.8	19.8	29.5	20.8	34.9	37.0
		15	15.4	21.3	29.9	22.8	35.2	36.9
		20	16.5	22.0	30.4	22.4	35.9	37.8

## Data Availability

Data is contained within the article.

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
