# Peer review of "FDTD Simulations of Shell Scattering in Au@SiO2 Core–Shell Nanorods with SERS Activity for Sensory Purposes"

_nanomaterials, 2022, doi:10.3390/nano12224011_

Round 1
Reviewer 1 Report
1. Authors are requested to revise according to the format published on the official website, which is a very detailed work, especially in the format of references and citations;
2. From the content of Abstract, this part belongs to the brief description area of the full text, and your specific future application prospects should be highlighted at the end;
3. In theoretical models, formula layout is very bad. If the authors have trouble with the formula processing, can they use Latex to edit. From ancient times to the present, a properly organized layout of papers is a respect for reviewers, and I believe that the champions of ancient China were not obtained by holding scribbles. At the same time, the font size in the formula is not uniform; There are also cases where Chinese and English punctuation marks are mixed in the formula;
4. In the FDTD approach, whether ∆x, ∆y, ∆z need to be italicized as in the simulation process;
5. In the Simulation process, there is a problem with the numbering of the table, and the table one appears here in the NR without SiO2 shell later;
6. In Simulation results, tables are laid out at their worst.
Author Response
Dear Reviewer,
Thank you for your interest and for your comments to our paper. We have carefully revised the manuscript and provide responses. Please find our answers in the attached file.
Best regards,
Authors

Reviewer 2 Report
According to the manuscript with title: " FDTD simulations of shell scattering in Au@ SiO2 core–shell nanorods with SERS activity for sensory purposes". The submitted work is introducing a new valuable and interesting idea and the given results confirm the idea. This work is suitable for publication in the Journal. I suggest the acceptance after some major corrections as follows;
1. Abstract section need to rewrite in correct sequence with more explanation
2. What is the application of this study ?
3. Reformulate the aim of the work in introduction
4. Add previous published work with comparison to clear the novelty of your work
5. Add critical results with numbers in abstract section
6. Give some results with numbers in conclusion
7. Some words are in cross-linked with others
8. The caption of all figures is very thin, need to add more details in each caption
9. Add more explanation to experimental work
10. Correct typographical errors.
11. Don't use abbreviations in title and abstract, you must define it in first time use
Author Response

(The authors gave the same response as above.)

Reviewer 3 Report
The author did excellent work. Accept the paper in current format
Author Response
Dear Reviewer,
Thank you for your interest and for your comments to our paper.
Best regards,
Authors
Round 2
Reviewer 2 Report
Accept